# Differences by sex and type of hypertension in mortality from hypertensive diseases between 1997 and 2020, and predictions for 2035 in Latin American and Caribbean countries

J. Smith Torres-Roman[1]*, Romina Barrera-Quispe[1], Carlos Quispe-Vicuña[2], Wagner Rios-Garcia[3], Bryan Rudas-Sulca[4], Yenko Damjanovic-Burga[4], Ivan Alegre-Cordero[4], Carito Zumaeta-Cabrera[2], Abdier Vizcarra-Tellez[5], Luis M. Tasayco-Márquez[6], Milward Ubillus[7], Carlo La Vecchia[8]

1 Universidad Científica del Sur, Lima, Peru, 2 Latin American Network for Cancer Research (LAN–CANCER), Lima, Peru, 3 Universidad Nacional San Luis Gonzaga de Ica, Ica, Peru, 4 Sociedad Cientifica de San Fernando, Universidad Nacional Mayor de San Marcos, Lima, Peru, 5 Division of Cardiology, Emergency Hospital of Villa El Salvador, Lima, Peru, 6 Escuela de Posgrado, Universidad Tecnologica del Peru, Lima, Peru, 7 Universidad de Huanuco, Huanuco, Peru, 8 Department of Clinical Sciences and Community Health, Università degli Study di Milano, Milan, Italy

* jstorresroman@gmail.com

## Abstract

### Background

Hypertension is the most prevalent cardiovascular condition and a leading contributor to premature mortality in Latin America and the Caribbean (LAC). Despite its public health relevance, few studies have examined long-term trends in hypertensive disease mortality across the region. The objective was to analyze trends in mortality due to hypertensive diseases from 1997 to 2020 in LAC countries, disaggregated by sex and type of hypertension, and to project mortality rates to 2035.

### Methods

We conducted an ecological time-series study using age-standardized mortality rates (ASMRs) extracted from the World Health Organization Mortality Database. Mortality from hypertensive diseases was defined by ICD-10 codes I10–I13 and further stratified into primary hypertension (I10) and hypertension-mediated organ damage (HMOD, I11–I13). Trends were assessed using Joinpoint regression to estimate average annual percent change (AAPC) and 95% confidence intervals (95%CI). Projections to 2035 were calculated using Norpred based on age – period – cohort analysis.

### Results

Between 1997 and 2020, mortality from hypertensive diseases increased in most LAC countries. Among men, significant increases were observed in Brazil

**Data availability statement:** The dataset underlying the findings of this study is publicly available in the Harvard Dataverse repository: https://doi.org/10.7910/DVN/PHWW6F. This includes the minimum data used for the analyses, such as country-level mortality counts and rates by sex and hypertension subtype. All methods are described in the paper.

**Funding:** The author(s) received no specific funding for this work.

**Competing interests:** The authors have declared that no competing interests exist.

(AAPC: 2.6%), El Salvador (6.6%), and Panama (6.1%), while Trinidad and Tobago was the only country showing a significant decline (AAPC: −5.7%). Among women, the most marked increase was seen in El Salvador (7.6%). By 2035, the Dominican Republic, Venezuela, and Paraguay are projected to have the highest ASMRs for hypertensive diseases among men, while for women, the Dominican Republic and Venezuela will remain among the most affected. Mortality from primary hypertension represented the greatest burden across countries, while HMOD exhibited lower but variable rates.

## Conclusion

Hypertensive disease mortality is projected to rise in LAC, with notable disparities by sex, country, and type of hypertension. These findings underscore the urgent need for regionally tailored prevention, screening, and control strategies.

## Introduction

Hypertension (HTN) is the most prevalent cardiovascular condition globally and represents a critical public health challenge [1,2]. According to the World Health Organization (WHO), the number of adults with hypertension doubled from 650 million in 1990 to 1.3 billion in 2019 [3]. Though, the estimated age-standardized prevalence of hypertension changed little between 1990 and 2019, increasing from 32% to 33%. In 2019, more than half of cardiovascular deaths were attributable to systolic hypertension, and 62% of deaths attributed to systolic hypertension occurred in adults aged 70 years and older [1].

Hypertension is a major risk factor for severe comorbidities, including kidney failure, stroke, and cardiovascular disease [4–9]. This highlights the urgency of implementing effective prevention, diagnosis, and control strategies. In 2019, in Latin American countries (LAC), [10], the prevalence of hypertension ranged widely between 18 and 62%, with the highest prevalence are Argentina, Brazil, Paraguay and Uruguay [11]. Its close association with diabetes and dyslipidemia further increases cardiovascular risk [12,13], as well as hypertension-mediated organ damage (HMOD) [14–16].

Advances in health and education in Latin America, combined with population aging, have contributed to an increasing burden of non-communicable diseases, with hypertension emerging as the leading cardiovascular risk factor [17]. These dynamics highlight the need to account for the region's unique epidemiological characteristics to optimize clinical management and reduce disease burden.

Despite the high burden of hypertension in LAC, studies on the evolution of mortality due to hypertensive diseases in the region remain limited. Deficiencies in knowledge, treatment, and disease control have contributed to high rates of premature mortality. In this context, the present study aims to analyze the trends in mortality rates due to hypertensive diseases in LAC between 1997 and 2020, disaggregated by sex and type of HTN. Furthermore, it provides mortality projections to the year

2035, offering a comprehensive view of current patterns and future trajectories that can inform regional cardiovascular health policy.

## Materials and methods

An ecological time-series study was conducted using the complete dataset available in the WHO Mortality Database.

Mortality data were extracted from the WHO mortality database, based on each country's official reporting systems. It is important to note that these deaths have not been adjusted to correct for possible underreporting or misclassification of deaths. Generally, these statistics are obtained using a mixed methodology, which integrates passive surveillance through national registration systems and active surveillance for case detection and confirmation. Although the integrity of these methods can vary significantly by country and year, in general these data accurately reflect the information presented by national health authorities. Therefore, the estimates used in this study seek to align closely with official statistics on mortality from HTN. Furthermore, the methodology addressed in this study is based on that used in previous publications [18,19].

Mortality data were collected for the period between 1997 (or the first available year) and 2020 (or the most recent available year) depending on data availability for each country. Deaths attributed to hypertensive diseases were identified using ICD-10 codes I10–I13. For analytical purposes, these were further classified into two subgroups: primary (essential) hypertension (I10) and hypertension-mediated organ damage (HMOD), which includes hypertensive heart disease (I11), hypertensive renal disease (I12), and hypertensive heart and renal disease combined (I13). This was done to differentiate risk factors between the onset and progression of the disease. It also allowed for a more detailed assessment of trends and differences in disease burden between subtypes, this distinction being critical for prevention and supported by evidence of hypertension as a promoter of risky atherosclerotic plaques [20]. The mortality database was organized into 18 five-year age groups (0–4, 5–9, 10–14,..., 85 + years), which were then used to calculate mortality rates by year.

### Statistical analysis

We estimated age-standardized mortality rates (ASMRs) per 100,000 person-years using the direct method, with the SEGI world standard population as the reference [21]. Trends in mortality rates were analyzed using the Joinpoint Regression Program, version 4.9.0 [22,23], which fits a series of joined straight lines (segments) to the logarithm of the mortality rates and identifies significant changes in trend (joinpoints) over time. The analysis employed the Monte Carlo permutation test with 4,999 permutations to determine the optimal number and location of joinpoints. An overall significance level of 0.05 was used to assess model fit. The analysis identified joinpoints and calculated the annual percent change (APC) along with corresponding 95% confidence intervals (95% CI) for each country. In cases where the model identified two or more segments, the average APC (AAPC) was computed as a weighted average of the APCs across those segments, based on the length of each interval. AAPCs were considered statistically significant when the p-value was < 0.05.

### Predictions through 2035

Mortality projections through 2035 were generated using the Nordpred package in R version 4.3.1. Nordpred is based on an age–period–cohort (APC) [24]. To prevent overestimation of long-term trends, attenuation of the linear drift component was applied, as recommended by the model's guidelines: 0% for the first projected period (2021–2025), 25% for the second period (2026–2030), and 50% for the third period (2031–2035) [24]. These projections were stratified by sex and type of hypertension, providing detailed insights into the expected burden of hypertensive disease mortality under current epidemiological patterns. In addition to estimating absolute numbers and age-standardized mortality rates (ASRs). The total change in projected deaths was decomposed into two components: (1) changes due to population structure (age and size), and (2) changes due to risk, defined here as the variation in disease incidence, case fatality, and health system performance, independent of demographic trends [25].

## Ethical Consideration

This study is based exclusively on freely available secondary data from the WHO Mortality Database. Since the dataset does not contain any personal identifiers, it was not necessary to obtain ethical approval for its use.

## Results

Among men in 2020, the highest mortality rates were observed in Nicaragua (47.1 per 100,000), Venezuela (35.3), and Ecuador (30.8). By 2035, the Dominican Republic (56.2), Venezuela (44.0), and Paraguay (34.8) are projected to have the highest rates (**Fig 1A**). Regarding primary hypertension, projections for 2035 indicate that the Dominican Republic (70.4), Paraguay (19.4), and Brazil (12.9) will experience the highest mortality rates (**Fig 1B**). In the case of HMOD, the highest projected mortality rates by 2035 will be reported in Nicaragua (52.4), Venezuela (36.4), and Panama (35.9) (**Fig 1C**). Among women, the highest age-standardized mortality rates from hypertensive diseases in 2020 were reported in the Dominican Republic (32.7 per 100,000), Nicaragua (23.0), and Venezuela (22.7). By 2035, the Dominican Republic (40.8) and Venezuela (27.4) are projected to continue having the highest mortality rates in the region (**Fig 1D**). In terms of primary hypertension, the Dominican Republic recorded the highest mortality rate in 2020 (27.2), and it is expected to remain the country with the highest burden by 2035 (52.3) (**Fig 1E**). Lastly, for HMOD, the highest projected mortality rates for 2035 are observed in Nicaragua (20.2), Cuba (18.4), and Panama (14.7) (**Fig 1F**).

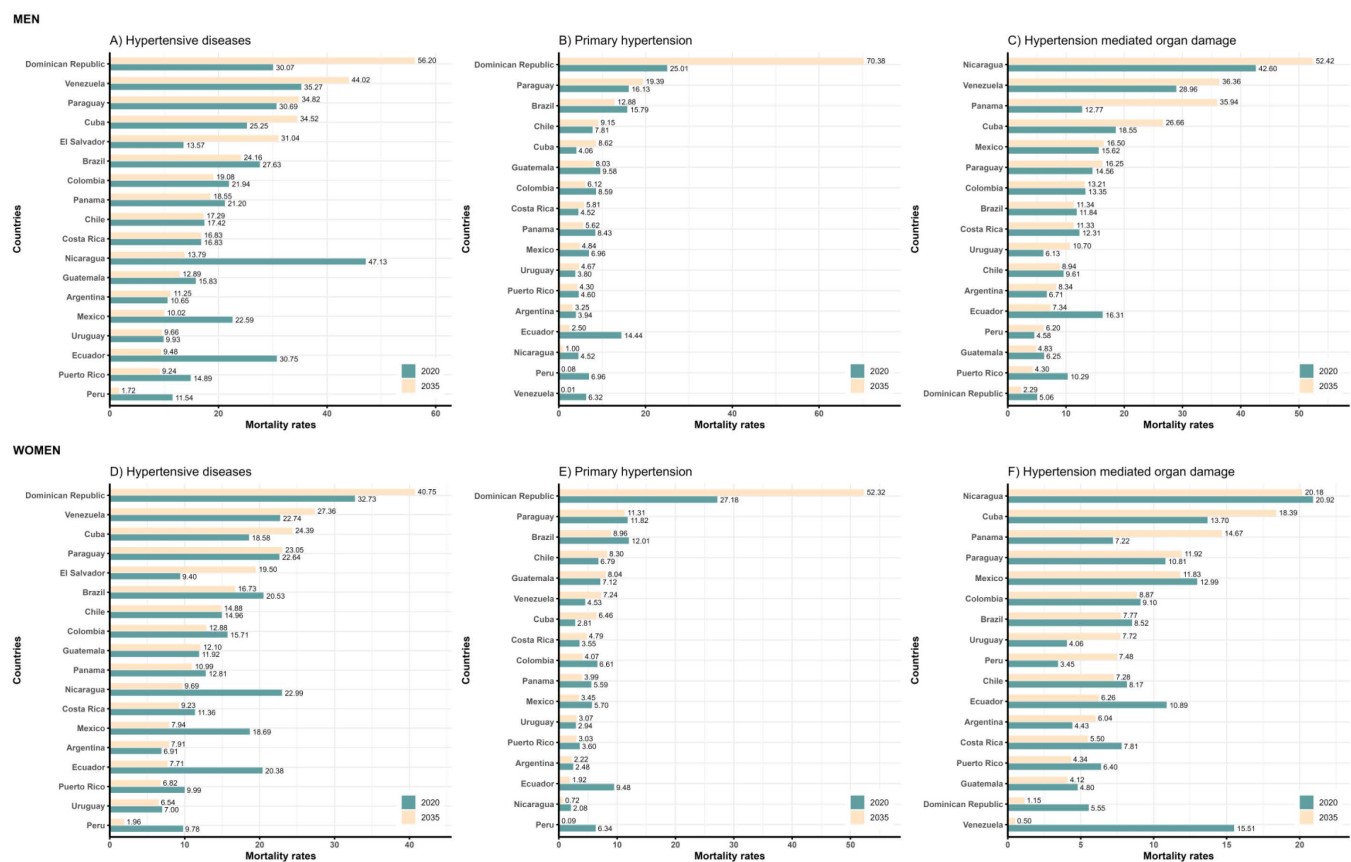

**Fig 1. Age-adjusted mortality rates by type of hypertension and sex for 2020 and their prediction for 2035.**

## Trends for hypertensive diseases

Among men of all ages, eight countries exhibited significant increases in hypertensive disease mortality: Brazil (2.6%), Cuba (5.1%), El Salvador (6.6%), Mexico (3.4%), Nicaragua (2.5%), Panama (6.1%), Paraguay (4.6%), and Venezuela (1.4%) exhibited significant increasing trends in mortality from hypertensive diseases. In contrast, Trinidad and Tobago experienced a significant decline of 5.7% between 1999 and 2012 (**Fig 2A and S1Table**). Among women, five countries experienced significant increases: Brazil (1.2%), Cuba (3.8%), Dominican Republic (2.6%), El Salvador (7.6%), and Paraguay (3.7%) also showed significant upward trends, while no country demonstrated a statistically significant decrease over the study period (**Fig 2D and S2 Table**).

## Trends for primary hypertension

For primary hypertensive mortality (I10), among men, Brazil (4.8%), Chile (3.2%), Colombia (2.9%), Costa Rica (7.2%), Cuba (2.5%), El Salvador (8.2%), Mexico (3.2%), Panama (7.8%), Paraguay (9.1%), and Trinidad and Tobago (4.0%)

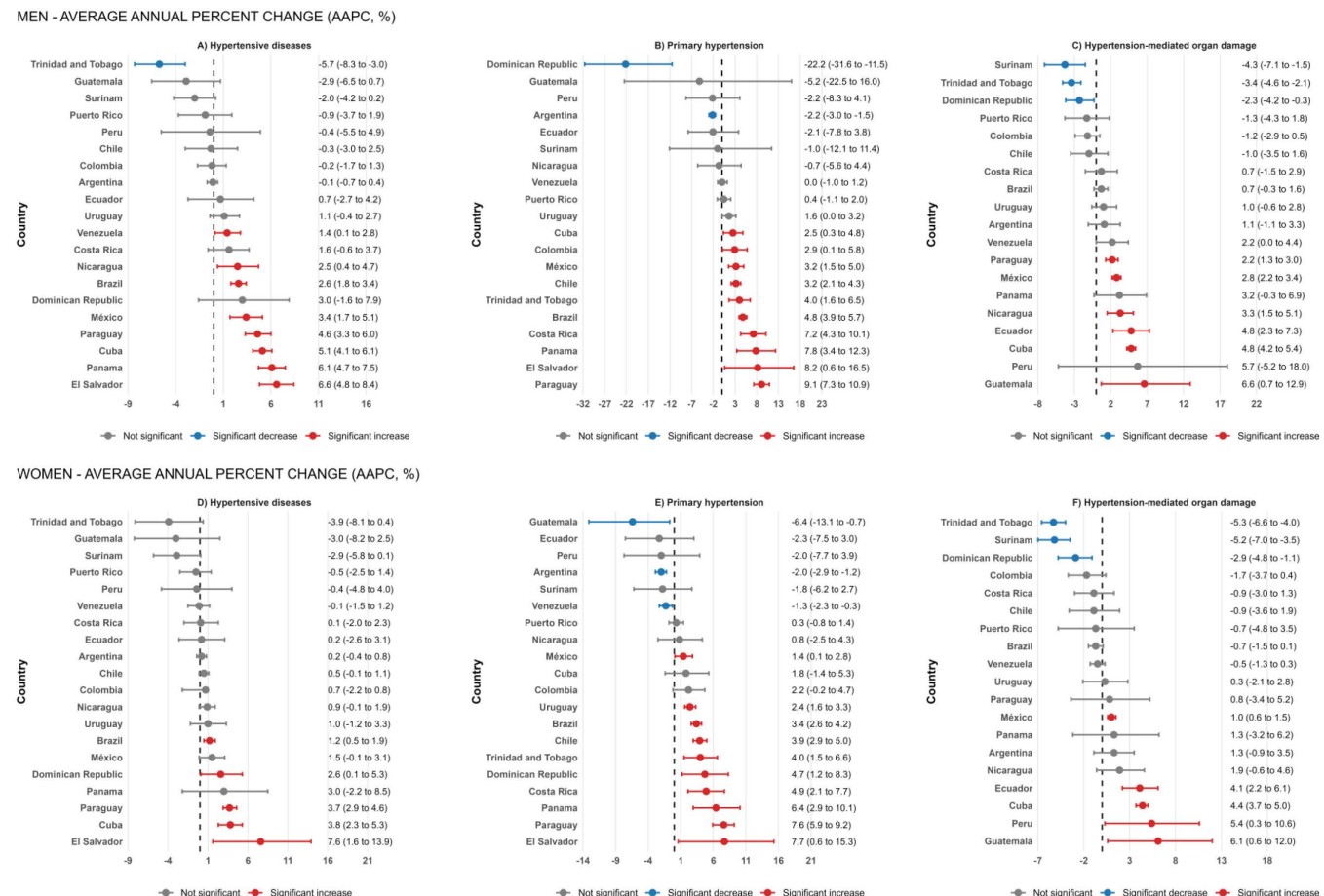

**Fig 2. Average annual percentage change in mortality rates from hypertensive diseases, primary hypertension, and organ damage-mediated hypertension, 1997-2020.**

(**Fig 2B** and **S3 Table**) Among women, Brazil (3.4%), Chile (3.9%), Costa Rica (4.9%), Dominican Republic (4.7%), El Salvador (7.7%), Mexico (1.4%), Panama (6.4%), Paraguay (7.6%), Uruguay (2.4%), and Trinidad and Tobago (4.0%). In contrast, Argentina (−2.0%), Guatemala (−6.4%), and Venezuela (−1.3%) experienced significant decreases (**Fig 2E** and **S4 Table**).

### Trends for hypertension-mediated organ damage (HMOD)

Among men, mortality from HMOD exhibited heterogeneous patterns across Latin American and Caribbean countries. Significant increases were observed in Cuba (4.8%), Ecuador (4.8%), Guatemala (6.6%), Mexico (2.8%), Nicaragua (3.3%), and Paraguay (2.2%). In contrast, the Dominican Republic (−2.3%), Surinam (−4.3%), and Trinidad and Tobago (−3.4%) showed significant decreasing trends in HMOD-related mortality (**Fig 2C** and **S5 Table**). Among women, Cuba (4.4%), Ecuador (4.1%), Guatemala (6.1%), Mexico (1.0%), and Peru (5.4%). On the other hand, the Dominican Republic (−2.9%), Surinam (−5.2%), Trinidad and Tobago (−5.3%) showed significant decreasing trends (**Fig 2F** and **S6 Table**).

Tables 1 and 2 show the estimated number of deaths from HTN, age-standardized mortality rates, and the percentage change in cases due to population growth and risk in Latin America and the Caribbean, 2020 and projected for 2035. Among men, notable disparities were observed in the projected changes in hypertensive disease mortality between 2020 and 2035 across Latin America and the Caribbean. The Dominican Republic showed one of the most alarming trends, with a 206.6% increase in the number of deaths and a sharp rise in the age-standardized mortality rate (ASMR), driven primarily by a 140.8% increase in risk-related mortality. Similarly, Cuba and El Salvador experienced substantial increases in ASMR and in the risk component (+108% and +172.4%, respectively), suggesting a worsening in disease control or growing exposure to key risk factors. In contrast, countries such as Peru and Puerto Rico demonstrated favorable

**Table 1. Number of hypertensive disease deaths, age-standardized mortality rates, and percentage change in cases due to population growth and risk among men in Latin America and the Caribbean, 2020 and predicted 2035.**

| Countries | Male population (annual million) | | Number of deaths in men | | Age-standardized mortality rates | | Change total (%) | Change due to population (%) | Change due to risk (%) |
|---|---|---|---|---|---|---|---|---|---|
| | 2020 | 2035 | 2020 | 2035 | 2020 | 2035 | | | |
| Argentina | 22 | 24.9 | 16987 | 25425 | 10.65 | 11.25 | 49.7 | 44 | 5.7 |
| Brazil | 103.3 | 111.1 | 129439 | 285510 | 27.63 | 24.16 | 120.6 | 125.1 | −4.5 |
| Chile | 9.3 | 9.8 | 12843 | 26166 | 17.42 | 17.29 | 103.7 | 86.9 | 16.8 |
| Colombia | 24.3 | 26.8 | 22849 | 57283 | 21.94 | 19.08 | 150.7 | 163.5 | −12.8 |
| Costa Rica | 2.5 | 2.8 | 2218 | 6321 | 16.83 | 16.83 | 185 | 140.4 | 44.6 |
| Cuba | 5.6 | 5.4 | 12014 | 31114 | 25.25 | 34.52 | 159 | 51 | 108 |
| Dominican Republic | 5.4 | 6 | 8689 | 26637 | 30.07 | 56.2 | 206.6 | 65.8 | 140.8 |
| Ecuador | 8.5 | 10.4 | 9146 | 9450 | 30.75 | 9.48 | 3.3 | 120.5 | −117.2 |
| El Salvador | 2.9 | 3.2 | 2036 | 6641 | 13.57 | 31.04 | 226.2 | 53.9 | 172.4 |
| Guatemala | 8.3 | 11.3 | 3238 | 7101 | 15.83 | 12.89 | 119.3 | 91.2 | 28.2 |
| Mexico | 60.6 | 71.2 | 20988 | 54932 | 22.59 | 10.02 | 161.7 | 63 | 98.7 |
| Nicaragua | 3.2 | 3.8 | 1200 | 3069 | 47.13 | 13.79 | 155.8 | 157.7 | −1.9 |
| Panama | 20.5 | 26 | 939 | 4997 | 21.2 | 18.55 | 432.2 | 124.8 | 307.4 |
| Paraguay | 3.2 | 4.2 | 3702 | 9795 | 30.69 | 34.82 | 164.6 | 11.3 | 53.3 |
| Peru | 15.9 | 18.5 | 6383 | 3312 | 11.54 | 1.72 | −48.1 | 94.5 | −142.6 |
| Puerto Rico | 1.6 | 1.3 | 3018 | 2521 | 14.89 | 9.24 | −16.5 | 25.6 | −42 |
| Uruguay | 1.7 | 1.8 | 1041 | 2114 | 9.93 | 9.66 | 103.1 | 35.9 | 67.2 |
| Venezuela | 15.2 | 17.1 | 19995 | 53765 | 35.27 | 44.02 | 168.9 | 115.5 | 53.4 |

**Table 2. Number of hypertensive disease deaths, age-standardized mortality rates, and percentage change in cases due to population growth and risk among women in Latin America and the Caribbean, 2020 and predicted 2035.**

| Countries | Female population (annual million) | | Number of deaths in men | | Age-standardized mortality rates | | Change total (%) | Change due to population (%) | Change due to risk (%) |
|---|---|---|---|---|---|---|---|---|---|
| | 2020 | 2035 | 2020 | 2035 | 2020 | 2035 | | | |
| Argentina | 22.4 | 25.9 | 22365 | 32681 | 6.91 | 7.91 | 46.1 | 33.6 | 12.5 |
| Brazil | 106.7 | 116.1 | 144839 | 307563 | 20.53 | 16.73 | 112.3 | 136.4 | −24.1 |
| Chile | 9.4 | 10 | 18364 | 34142 | 14.96 | 14.88 | 85.9 | 70.1 | 15.8 |
| Colombia | 24.9 | 27.8 | 24331 | 53889 | 15.71 | 12.88 | 121.5 | 155.4 | −33.9 |
| Costa Rica | 2.5 | 2.8 | 2224 | 4709 | 11.36 | 9.23 | 111.8 | 139.7 | −28 |
| Cuba | 5.6 | 5.5 | 11696 | 29756 | 18.58 | 24.39 | 154.4 | 59.1 | 95.3 |
| Dominican Republic | 5.4 | 6 | 8160 | 26632 | 32.73 | 40.75 | 226.4 | 76.1 | 150.3 |
| Ecuador | 8.5 | 10.4 | 9480 | 10282 | 20.38 | 7.71 | 8.5 | 110.7 | −102.3 |
| El Salvador | 3.3 | 3.7 | 2246 | 7277 | 9.4 | 19.5 | 224.0 | 75.6 | 148.4 |
| Guatemala | 8.5 | 11.5 | 3759 | 8888 | 11.92 | 12.1 | 136.5 | 104.7 | 31.7 |
| Mexico | 63.2 | 74.5 | 28495 | 61262 | 18.69 | 7.94 | 115 | 76.8 | 38.2 |
| Nicaragua | 3.3 | 4 | 1345 | 3474 | 22.99 | 9.69 | 158.3 | 164.8 | −6.5 |
| Panama | 2.1 | 2.6 | 884 | 3844 | 12.81 | 10.99 | 334.9 | 133.1 | 201.8 |
| Paraguay | 3.2 | 4.1 | 3684 | 8229 | 22.64 | 23.05 | 123.4 | 99.8 | 23.6 |
| Peru | 16.2 | 18.7 | 7086 | 5133 | 9.78 | 1.96 | −27.6 | 111.2 | −138.8 |
| Puerto Rico | 1.8 | 1.5 | 3397 | 4080 | 9.99 | 6.82 | 20.1 | 57.3 | −37.2 |
| Uruguay | 1.7 | 1.8 | 1543 | 2748 | 7 | 6.54 | 78.1 | 20.2 | 58 |
| Venezuela | 15.4 | 17.7 | 19180 | 49291 | 22.74 | 27.36 | 157 | 122.6 | 34.4 |

trends. Peru showed a 48.1% reduction in total deaths and a dramatic decline in ASMR (from 11.5 to 1.7), driven by a 142.6% reduction in risk, despite an increase in population size. Puerto Rico also reported a moderate decrease in deaths (−16.5%) and a 42% reduction in risk-related mortality (**Table 1**). Among women, projections reveal substantial variation across countries in hypertensive disease mortality by 2035. While most countries are expected to experience an increase in the total number of deaths due to demographic expansion, several show diverging trends in age-standardized mortality rates (ASMRs), highlighting underlying changes in risk. The Dominican Republic presents one of the most concerning profiles, with a 226.4% increase in deaths and a notable rise in ASMR (from 32.7 to 40.8), largely driven by a 150.3% increase in the risk component, signaling a worsening in disease control and exposure. Similarly, Cuba and El Salvador show high increases in both total deaths and ASMRs, with large risk-related components (+95.3% and +148.4%, respectively). In contrast, countries such as Peru, Puerto Rico, and Colombia demonstrate encouraging trends. Peru stands out with a 27.6% reduction in total deaths and a steep drop in ASMR (from 9.8 to 2.0), reflecting a 138.8% reduction in risk, despite population growth. Puerto Rico also shows a reduction in ASMR and a risk decrease of −37.2%. Colombia, while experiencing a rise in total deaths due to demographics, shows a substantial risk reduction (−33.9%), suggesting improvements in hypertension control (**Table 2**).

Finally, the projections for deaths, age-standardized mortality rates (ASMRs), and contributions of population growth and disease risk to the change in burden from primary hypertension (I10) and hypertension-mediated organ damage (HMOD, I11–I13) among men and women in Latin America and the Caribbean by 2035 (**S7 Table** to **S10 table**). Notably, countries like the Dominican Republic, Paraguay, and Cuba are projected to experience substantial increases in deaths from primary hypertension, particularly among men, with increases primarily driven by rising risk rather than demographic changes. In contrast, countries such as Peru, Ecuador, and Nicaragua are expected to see dramatic reductions in both

deaths and ASMRs, reflecting possible improvements in risk factor control or data inconsistencies (**S7 Table** and **S8 Table**). For HMOD, the projections suggest alarming increases in countries like Nicaragua, Panama, and Peru, especially among women, where risk-related change explains most of the predicted burden. Meanwhile, countries such as the Dominican Republic, Ecuador, and Venezuela show projected reductions in HMOD mortality, but often with unclear or potentially misleading underlying trends (**S9 Table** and **S10 Table**).

Across all categories of hypertensive disease, primary hypertension (I10), and hypertension-mediated organ damage (HMOD, I11–I13), mortality was consistently higher among men than women in Latin America and the Caribbean. This sex disparity was evident throughout the study period and across most countries, highlighting a persistent and disproportionate burden of hypertensive mortality in male populations.

## Discussion

This study provides an updated and comprehensive assessment of mortality trends from hypertensive diseases in 19 countries across LAC between 1997 and 2020, along with projections to 2035. Overall, the persistent or rising mortality rates observed in several countries, despite advances in hypertension awareness and treatment globally, underscore the limited effectiveness of current prevention and control strategies in LAC. The observed sex-based disparities further highlight the need to integrate gender-sensitive approaches into cardiovascular prevention programs. Moreover, the divergence between trends in primary hypertension and HMOD suggests variations in diagnostic coding, comorbidity burden, and healthcare quality across settings.

By including long-term projections, this study provides valuable foresight into the potential trajectory of hypertension-related mortality, emphasizing the urgency of strengthening primary prevention, improving detection and adherence to therapy, and addressing social determinants of cardiovascular health. These findings should inform national and regional policies aimed at reversing the growing burden of hypertension in the region.

Our findings are broadly consistent with estimates from the Global Burden of Disease (GBD), which identify cardiovascular disease (CVD) as the leading cause of death worldwide and hypertension as one of the main modifiable risk factors contributing to premature mortality [2,26,27]. Globally, the age-standardized prevalence of hypertension increased slightly between 1990 and 2019. While high-income countries and the WHO European Region saw a decline in prevalence, other regions, including the WHO Western Pacific Region (from 24% to 28%) and the WHO South-East Asia Region, experienced modest increases. Nevertheless, the absolute number of adults aged 30–79 with elevated blood pressure continues to rise, largely due to population aging and demographic expansion [1]. In the Americas, the age-standardized prevalence of CVD in 2021 was estimated at 7.7%, lower than in the Eastern Mediterranean Region (10.1%). However, the absolute number of CVD cases has increased, driven by demographic shifts, despite declining age-standardized rates. In 2021, CVD accounted for 11% of all cardiovascular deaths globally. Although the Americas represent a relatively smaller share of global CVD deaths, cardiovascular disease is responsible for one-third of all deaths in the region [28]. In recent decades, the region has undergone rapid and complex epidemiological transitions, characterized by a rising burden of non-communicable diseases (NCDs). Population aging, urbanization, and lifestyle change are the primary drivers of the growing impact of NCDs across Latin America and the Caribbean. In fact, the population of Latin America and the Caribbean tripled between 1950 and 2010, rising from 167 million to 588 million, representing 8.5% of the global population [29].

In LAC, the analyses by sex revealed significant differences. Among men, increases were recorded in eight countries, whereas Trinidad and Tobago showed a decrease by 5.7% annually. In contrast, among women, significant upward trends were recorded in five countries, and no country exhibited a significant decline. This can be explained by biological causes, such as the female protective factor before menopause; however, afterwards, both men and women would have similar patterns of risk [30]. For this reason, we consider that the main cause must be different, such as treatment adherence, which is crucial to reducing mortality [31,32]. A study carried out in four South American cities reveals that women have a higher association of their disease with hypertension (36.1% of men and

62.1% of women among patients with hypertension) and are also more likely to be treated (38.1% of men and 46.5% of women) [33]. The social determinants of health are crucial to understanding this, since in Latin America there remains a certain form of gender discrimination that prevents the acceptance of illness; thus, various social determinants of health can be considered in this context [34,35].

When analyzing by subtype, countries such as Paraguay, Panama, El Salvador, Brazil, Trinidad and Tobago, Chile, and Mexico exhibited increasing mortality from primary hypertension. Notably, Brazil experienced a sharp rise in mortality between 1997 and 2008, followed by a significant decline until 2018, and then a renewed increase from 2018 to 2020. This recent reversal if not partly due to changes in certifications, may reflect disruptions in the health system and the indirect impact of crises such as the COVID-19 pandemic (26). Countries like Costa Rica also exhibited abrupt reversals in mortality trends, possibly associated with fluctuations in health system performance, data quality, demographic shifts, or broader epidemiological transition (24).

Regarding HMOD, countries such as Cuba, Ecuador, Guatemala, Mexico, and Peru showed increasing mortality trends in both men and women, suggesting a progression toward more advanced or poorly controlled stages of hypertension. These trends are particularly concerning given that HMOD is a strong predictor of cardiovascular mortality and adverse clinical outcomes (27). The observed increases may be attributed not only to the rising prevalence of hypertension but also to poor disease control and structural deficiencies in healthcare systems. For example, in Cuba, high average systolic blood pressure combined with the coexistence of obesity and diabetes has been reported to accelerate progression to organ damage [36]. In Guatemala, significant infrastructure limitations have been documented, including an inconsistent supply of antihypertensive medications [37]. In Ecuador, only 49% of individuals with hypertension were aware of their condition, 40% received treatment, and just 19% had controlled blood pressure [38]. Similarly in Mexico, disease control remained low between 2000 and 2019, with only half of the patients adhering to prescribed therapy [39]. In Peru, awareness and control of hypertension have been alarmingly low, with only around 5% of hypertensive patients achieving adequate blood pressure control, and a decline in awareness reported in recent years [40]. Taken together, these findings suggest that the increasing burden of HMOD in the region is not only a consequence of the rising prevalence of hypertension, but also of persistent barriers to access, low treatment adherence, and structural inequalities that hinder effective and timely management.

By 2035, global and primary hypertension mortality trends are projected to decline, contrasting with the APCs observed in some Latin American countries. However, in the case of HMOD, a sustained increase is expected. This can be explained by the fact that, although the diagnosis and treatment of hypertension have improved, problems persist in adequately controlling blood pressure, with almost all regions in LAC reporting that less than half of their hypertensive population has their blood pressure under control [41]. In addition, population aging [42] and the high prevalence of factors such as obesity (62%) [43] or diabetes [44] maintain a high risk of organ damage, even in treated patients. Thus, while mortality from uncomplicated hypertension may stabilize or decline, mortality associated with HMOD will continue to increase due to the cumulative weight of risk factors.

In Peru, the downward trend in mortality due to high blood pressure, is supported by previous studies, where an appreciated decrease in the mortality rate due to hypertension was found, especially in the coastal region, where there is greater health system coverage and population concentration [45]. In addition a comparison made in 2022 of the causes of deaths between 2010 and 2018 with MINSA data, exhibit a decrease in the percentage of cause of death for hypertensive diseases (I10-I15) from 3.9% in 2010 to 1.8% in 2018 [46]. However, both studies present the similar limitations regarding the quality of the database employed. Notably, a marked increase in hypertension-related mortality was observed in the Peruvian population during the 2018–2020 period, a pattern that has also been reported in other countries. This abrupt rise coincides with the onset of the COVID-19 pandemic and may reflect its substantial impact on health systems, which led to reduced access to medical care, disruptions in the management of chronic conditions, and delays in cardiovascular disease control. [47] Therefore, it is likely that the observed increase is largely attributable to the indirect effects of the pandemic on health service delivery and hypertension management.

Mortality from primary hypertension represented the greatest burden across countries, with marked projected increases in nations such as the Dominican Republic. Although deaths attributed to secondary hypertension were less frequent, this may partly reflect the ICD-10 coding system used in mortality databases, where some cases of hypertension-mediated organ damage (HMOD) such as hypertensive heart or renal disease can be classified under secondary hypertension. This overlap could explain part of the variation observed among countries.

The present study shows substantial increases in the absolute number of deaths attributed to hypertension projected for 2035, in both men and women. However, age-standardized mortality rates present a more complex picture: while countries such as Peru, Ecuador, Nicaragua, Puerto Rico, and Mexico show a significant decrease in mortality, other countries, such as Venezuela, the Dominican Republic, Paraguay, Cuba, and El Salvador, show a significant increase, suggesting a greater risk at the population level, beyond population growth. This is consistent with global and regional reports highlighting the combined impact of aging, urbanization, dietary changes, and inequalities in health systems [48,49]. In light of these findings, there is an urgent need for sustained, regionally tailored hypertension control strategies that include robust primary prevention, universal access to early detection and effective treatment, and targeted interventions for high-risk groups. Strengthening health system resilience is essential to avoid care disruptions during crises, while improving epidemiological surveillance, standardizing diagnostic criteria, and enhancing mortality data quality will be critical for guiding evidence-based policy. Without such comprehensive efforts, the projected rise in hypertension-related mortality could reverse public health gains achieved in the region over recent decades.

## Limitations

This study has some limitations. First, the accuracy of mortality data depends on the quality of death certification and cause-of-death coding, which may vary across countries and over time. Misclassification of causes, particularly between primary hypertension, HMOD, and related cardiovascular conditions may have led to under or overestimation in several settings and biased trends over time. Second, the ecological study design does not allow for causal inference or individual-level analysis of risk factors, treatments, or access to care. Lastly, while Nordpred projections are based on rigorous modeling, they assume that past trends will continue, and may not fully capture the impact of future interventions, policy changes, or unforeseen events such as pandemics. In contrast, the study also has several strengths. It is the first analysis in Latin America and the Caribbean to examine long-term trends and projections of mortality from hypertensive diseases, disaggregated by sex and by type of hypertension. The use of high-quality, publicly available WHO mortality data and standardized analytical methods enhances the comparability and robustness of the findings. Moreover, the inclusion of risk–population decomposition provides a more comprehensive understanding of the drivers behind mortality changes in the region.

## Conclusions

This study reveals a persistent and rising burden of hypertensive disease mortality in Latin America and the Caribbean, with considerable variation by sex, country, and hypertension subtype. The projected increases in mortality rates and absolute number of deaths by 2035 underscore the urgency of implementing regionally tailored prevention and control strategies. Strengthening health systems, expanding access to care, and improving epidemiological data quality will be key to mitigating the impact of HTN and preserving the gains made in cardiovascular health across the region.

## Supporting information

**S1 Fig. Trends of mortality rates from hypertension diseases in Latin American and the Caribbean population, 1997–2020.**
(TIFF)

**S2 Fig. Trends of mortality rates from primary hypertension in Latin American and the Caribbean population, 1997–2020.**
(TIFF)

**S3 Fig. Trends of mortality rates from organ damage-mediated hypertension in Latin American and the Caribbean population, 1997–2020.**
(TIFF)

**S1 Table. Annual percentage change in mortality trends from hypertensive diseases (I10-I13) for men in twenty countries in Latin America and the Caribbean, 1997–2020.**
(DOCX)

**S2 Table. Annual percentage change in mortality trends from hypertensive diseases (I10-I13) for women in twenty countries in Latin America and the Caribbean, 1997–2020.**
(DOCX)

**S3 Table. Average annual percent change and 95% confidence intervals for primary hypertension (I10) for men in twenty countries in Latin America and the Caribbean, 1997–2020.**
(DOCX)

**S4 Table. Average annual percent change and 95% confidence intervals for primary hypertension (I10) for women in twenty countries in Latin America and the Caribbean, 1997–2020.**
(DOCX)

**S5 Table. Average annual percent change and 95% confidence intervals for hypertension-mediated organ damage (I11-I13) for men in twenty countries in Latin America and the Caribbean, 1997–2020.**
(DOCX)

**S6 Table. Average annual percent change and 95% confidence intervals for hypertension-mediated organ damage (I11-I13) for women in twenty countries in Latin America and the Caribbean, 1997–2020.**
(DOCX)

**S7 Table. Number of primary hypertension (I10) deaths, age-standardized mortality rates, and percentage change in cases due to population growth and risk among men in Latin America and the Caribbean, 2020 and predicted 2035.**
(DOCX)

**S8 Table. Number of primary hypertension (I10) deaths, age-standardized mortality rates, and percentage change in cases due to population growth and risk among women in Latin America and the Caribbean, 2020 and predicted 2035.**
(DOCX)

**S9 Table. Number of hypertension-mediated organ damage (HMOD) (I11-I13) deaths, age-standardized mortality rates, and percentage change in cases due to population growth and risk among men in Latin America and the Caribbean, 2020 and predicted 2035.**
(DOCX)

**S10 Table. Number of hypertension-mediated organ damage (HMOD) (I11-I13) deaths, age-standardized mortality rates, and percentage change in cases due to population growth and risk among women in Latin America and the Caribbean, 2020 and predicted 2035.**
(DOCX)

**S1 File. STROBE Statement—checklist of items that should be included in reports of observational studies.**
(PDF)

## Acknowledgments

The authors thank the Universidad Científica del Sur for their support in the publication of this research.

## Author contributions

**Conceptualization:** J Smith Torres-Roman.

**Data curation:** J Smith Torres-Roman, Luis M. Tasayco-Márquez.

**Formal analysis:** Carlos Quispe-Vicuña.

**Methodology:** J Smith Torres-Roman, Carlos Quispe-Vicuña.

**Supervision:** Carito Zumaeta-Cabrera, Abdier Vizcarra-Tellez, Carlo La Vecchia.

**Validation:** J Smith Torres-Roman.

**Visualization:** Luis M. Tasayco-Márquez.

**Writing – original draft:** J Smith Torres-Roman, Romina Barrera-Quispe, Carlos Quispe-Vicuña, Wagner Rios-Garcia, Bryan Rudas-Sulca, Yenko Damjanovic-Burga, Ivan Alegre-Cordero, Carito Zumaeta-Cabrera, Abdier Vizcarra-Tellez, Luis M. Tasayco-Márquez, Milward Ubillus, Carlo La Vecchia.

**Writing – review & editing:** J Smith Torres-Roman, Romina Barrera-Quispe, Carlos Quispe-Vicuña, Wagner Rios-Garcia, Bryan Rudas-Sulca, Yenko Damjanovic-Burga, Ivan Alegre-Cordero, Carito Zumaeta-Cabrera, Luis M. Tasayco-Márquez, Milward Ubillus, Carlo La Vecchia.

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
