## [Decision Letter · Decision Letter 0]

30 Nov 2025

Dear Dr. Torres-Roman,

Thank you for submitting your manuscript to PLOS ONE. After careful consideration, we feel that it has merit but does not fully meet PLOS ONE’s publication criteria as it currently stands. Therefore, we invite you to submit a revised version of the manuscript that addresses the points raised during the review process.

We look forward to receiving your revised manuscript.

Kind regards,

Andres Mauricio Acevedo-Melo, M.D.

Academic Editor

PLOS ONE

Journal Requirements:

**Additional Editor Comments:**

In this manuscript, Torres-Román et al. present the results of an ecological study analyzing mortality trends from hypertensive diseases across 20 Latin American and Caribbean countries. Using WHO mortality database records from 1997 to 2020, the authors estimated age-standardized mortality rates (ASMR) by sex for each country and identified notable disparities in the average annual percentage change (AAPC). Additionally, the authors projected mortality rates to 2035 and reported an expected increase in AAPC for the region, underscoring the need for strengthened public health policies and tailored prevention strategies.

While the topic is relevant and contributes to understanding the regional burden of hypertensive disease, the manuscript lacks a sufficiently detailed methodological description, particularly regarding the stepwise estimation and modeling procedures used to calculate the ASMR. The manuscript also does not address the potential limitations associated with the reliability of mortality data, which may be underestimated due to incomplete death certificate reporting in several countries of the region. Furthermore, the analysis does not incorporate relevant contextual factors—such as socioeconomic indicators or public health system characteristics—that could influence or modify the observed trends and projections.

The authors are encouraged to submit a revised version that addresses these methodological and contextual issues. They are also required to adhere to an appropriate reporting guideline for ecological observational studies.

Reviewers' comments:

Reviewer's Responses to Questions

**Comments to the Author**

1. Is the manuscript technically sound, and do the data support the conclusions?

Reviewer #1: Partly

Reviewer #2: Yes

Reviewer #3: Partly

2. Has the statistical analysis been performed appropriately and rigorously?

Reviewer #1: No

Reviewer #2: Yes

Reviewer #3: No

3. Have the authors made all data underlying the findings in their manuscript fully available?

Reviewer #1: No

Reviewer #2: Yes

Reviewer #3: No

4. Is the manuscript presented in an intelligible fashion and written in standard English?

Reviewer #1: Yes

Reviewer #2: Yes

Reviewer #3: Yes

Reviewer #1: Please find the comments attached in a word document, as the system does not allow to upload the comments in this word count system, thanks for your understanding and sorry for the inconvenience. Have a wonderful day.

Reviewer #2: Interesting paper. Some issues should be added

I have no competing interests

Method section is too short. first authors should add if similar publication from WHO dataset is known and if yes quote it

Moreover regarding prediction authors should be more precise about it, and also add ways of evaluating quality of it

why authors separated primary and hypertension with organe damage?

hyperrension has been related to at risk plaques. plese quote on PMID: 26508517

Reviewer #3: In the conclusion of the summary, the researchers highlight notable disparities; however, they do not provide a comprehensive analysis of these differences. They employ a joinpoint model to illustrate the percentage change in mortality from hypertensive diseases across 18 Latin American countries. The forest plot graphs require improvements in their visual presentation, and an analysis of disparities by sex and by country should be incorporated. Furthermore, the study could benefit from examining how countries perform in relation to the Human Development Index (HDI) or Gross Domestic Product (GDP) and their respective mortality rates from hypertensive diseases.

**Do you want your identity to be public for this peer review?** For information about this choice, including consent withdrawal, please see our Privacy Policy

Reviewer #1: **Yes:** Henry Becerra

Reviewer #2: **Yes:** Fabrizio D'Ascenzo

Reviewer #3: No

---

## [Author Response · Author response to Decision Letter 1]

15 Dec 2025

15 December, 2025

Emily Chenette

Senior Editor of Plos One

Reference: PONE-D-25-57467

Dear Dra. Emily Chenette

We greatly appreciate the consideration given to our manuscript and the valuable comments made by the reviewers. Below we present a point-by-point response to the comments. As requested, we have highlighted the changes that were made from the previous version in the attached revised document.

We are grateful for the opportunity to publish our research work in your new journal. All co-authors have seen and agree with this resubmission.

Thank you for your consideration of our manuscript.

Sincerely,

J. Smith Torres Román, MD

Universidad Científica del Sur, Lima, Peru

Tel: +(51) 99 321-9508

Email: jstorresroman@gmail.com

Additional Editor Comments:

In this manuscript, Torres-Román et al. present the results of an ecological study analyzing mortality trends from hypertensive diseases across 20 Latin American and Caribbean countries. Using WHO mortality database records from 1997 to 2020, the authors estimated age-standardized mortality rates (ASMR) by sex for each country and identified notable disparities in the average annual percentage change (AAPC). Additionally, the authors projected mortality rates to 2035 and reported an expected increase in AAPC for the region, underscoring the need for strengthened public health policies and tailored prevention strategies.

While the topic is relevant and contributes to understanding the regional burden of hypertensive disease, the manuscript lacks a sufficiently detailed methodological description, particularly regarding the stepwise estimation and modeling procedures used to calculate the ASMR. The manuscript also does not address the potential limitations associated with the reliability of mortality data, which may be underestimated due to incomplete death certificate reporting in several countries of the region. Furthermore, the analysis does not incorporate relevant contextual factors—such as socioeconomic indicators or public health system characteristics—that could influence or modify the observed trends and projections.

The authors are encouraged to submit a revised version that addresses these methodological and contextual issues. They are also required to adhere to an appropriate reporting guideline for ecological observational studies.

Authors: We thank the reviewer for highlighting key aspects to improve. In the revised manuscript, we expanded the Methods section to include a more detailed description of the procedures used to calculate age-standardized mortality rates (ASMRs), including the use of the direct method with the SEGI world standard population and the five-year age group structure applied. We also clarified the modeling procedures used in Joinpoint regression and Nordpred projections. To address concerns about data quality, we added a discussion of potential limitations associated with the completeness and accuracy of mortality records across countries, including possible underreporting or misclassification in vital registration systems. Although our ecological design did not allow for the statistical inclusion of contextual variables, we expanded the Discussion to reflect how factors such as health system capacity, medication access, treatment adherence, and socioeconomic inequalities may have influenced the observed patterns. Finally, we revised the manuscript to follow the STROBE guidelines for observational studies, and we have included the completed checklist as a supplementary file.

Comments

Unify the typing of “Peru” and “Perú” in the affiliations

Keywords: Please use all in English

Authors: Done

In the conclusion of the summary, the researchers highlight notable disparities; however, they do not provide a comprehensive analysis of these differences. They employ a joinpoint model to illustrate the percentage change in mortality from hypertensive diseases across 18 Latin American countries. The forest plot graphs require improvements in their visual presentation, and an analysis of disparities by sex and by country should be incorporated. Furthermore, the study could benefit from examining how countries perform in relation to the Human Development Index (HDI) or Gross Domestic Product (GDP) and their respective mortality rates from hypertensive diseases.

Authors: We thank the reviewer for this valuable suggestion. In response, we would like to clarify that Figure 2 already presents the average annual percent change (AAPC) in mortality rates for each country, disaggregated by sex and further stratified by type of hypertensive disease—namely, primary hypertension (I10) and hypertension-mediated organ damage (HMOD, I11–I13). This multipanel forest plot was designed specifically to highlight disparities across countries and between sexes in a clear and comparative manner. Each subplot in Figure 2 (A to F) corresponds to a specific dimension (overall, primary, and HMOD), ensuring that the differences in trends are not only statistically reported but also visually accessible. We have improved the visual quality and labeling of the figure in the revised manuscript to enhance readability and ensure that these distinctions are fully appreciated.

Introduction

Introduction, paragraph 2, line 4: “(LAC), 2019, the prevalence ” connector seems to be missing

Introduction, paragraph 2, line 7: “12),as well” space is needed

Introduction, paragraph 3, line 1: “growth of the elderly population, have led to a growing” can improve writting

Introduction, paragraph 3, line 3: “(16).These “ space is needed

Authors: Thank you for your suggestions. The changes have been made.

Materials and methods

paragraphs 2, line 2: “2020 (or the most recent available year)” please clarify thatñ

Authors: We thank the reviewer for these thoughtful comments. In the revised Methods section, we have clarified that the phrase “1997 (or the first available year) to 2020 (or the most recent available year)” reflects the reality that not all countries in the WHO Mortality Database have complete time series spanning the full period.

paragraphs 2, line 9: “Mortality data were analyzed in five-year age intervals (0–4, 5–9, 10–14, ..., 85+ years).” this is not shown in the results

Authors: In the original text, the mention of five-year age intervals may have led to confusion regarding age-stratified analyses. To address this, we have revised the Methods section to clarify that these age groups (0–4 to 85+ years) were used solely for the calculation of age-standardized mortality rates (ASMRs) using the direct standardization method and the SEGI world standard population as the reference. We did not conduct or report age-specific analyses in the Results section. This clarification has been incorporated to ensure accuracy and avoid misinterpretation of the methodology.

Predictions through 2035, line 11: “disease risk” It would be great to elaborate more on how you estimated variation in disease risk.

Authors: We thank the reviewer for this observation. In the revised Methods section, we have expanded the explanation of how variation in “disease risk” was estimated using the Nordpred model, which is based on an age–period–cohort (APC) framework.

Method section is too short. first authors should add if similar publication from WHO dataset is known and if yes quote it. Moreover regarding prediction authors should be more precise about it, and also add ways of evaluating quality of it

why authors separated primary and hypertension with organe damage? hyperrension has been related to at risk plaques. plese quote on PMID: 26508517

Authors: We appreciate the reviewer’s comments regarding the need for greater methodological depth. In the revised Methods and Discussion sections, we now reference similar ecological studies that have used the WHO Mortality Database to analyze trends in cardiovascular or hypertensive disease mortality in Latin America

Results

paragraph 2, line 4: “ Trinidad and Tobago (AAPC: −5.7%) showed a significant decreasing trend” why is that? This is theoretically unexpected, and no mention of this finding is found in results.

Authors: We thank the reviewer for this important observation. Trinidad and Tobago was indeed the only country in our analysis showing a statistically significant decreasing trend in hypertensive disease mortality among men (AAPC: −5.7%). While this finding may seem unexpected given the regional trends, previous studies have reported relatively high antihypertensive treatment coverage and control rates in Trinidad and Tobago compared to other countries in the Caribbean. We have explained this in discussion.

paragraph 2, line 7: “ Republica Dominicana” need to unify the use of the name in the text and in the graph

Authors: We thank the reviewer for pointing out this inconsistency. We have revised the manuscript and figures to ensure consistent use of the country name “Dominican Republic” throughout the text, tables, and graphs, following standardized English-language country naming conventions.

paragraph 3, line 1: “ mortality trends” it would be great to show the graph trends

Authors: Thank you for the suggestion. We agree that including the mortality trends graph would enhance the presentation of the results. We will generate and provide the corresponding trend figures, which will be included as supplementary material in order to avoid exceeding the maximum number of tables and figures allowed by the journal.

paragraph 3, Figures and supplements: the figure does not seem to represent what is stated in the text, and the supplements are not available for review

Authors: Thank you for your observation. We have revised the text to ensure that it accurately reflects the results presented in the figures. Additionally, the supplementary material has been corrected and is now included in the submission for your review.

paragraph 4, line 5: “This sequence -if not largely influenced by changes in certification- suggests a significant structural shift in the disease’s trajectory, potentially influenced by changes in access to treatment, health policy adjustments, or broader socioeconomic factors.” and line 11 “These fluctuations underscores the instability of mortality trends and highlight in the country and the need for sustained public health interventions” : this is part of the discussion, not the results

Authors: Thank you for pointing this out. We agree with your observation that these sentences correspond to the discussion rather than the results.

paragraph 4, lines 13 and 22: the figure does not seem to represent what is stated in the text, and the supplements are not available for review

Authors: Thank you for your observation. We have revised the text to ensure that it accurately reflects the results presented in the figures. Additionally, the supplementary material has been corrected and is now included in the submission for your review.

last paragraph, line 1: “HTA” usually in english hypertension is abbreviated as HTN, and also need to show where was used abbreviation before (unable to find).

Authors: We thank the reviewer for this observation. We have revised the manuscript to replace the abbreviation “HTA” with the standard English abbreviation “HTN” for hypertension

last paragraph, line 2: “risk” how was it defined?

Authors: We appreciate the reviewer’s request for clarification. In the revised Methods section, we have now specified that within the Nordpred model, the term “risk” refers to the non-demographic component of the change in mortality rates, which captures the effect of changes in disease incidence, case fatality, and health system performance over time. This is distinct from changes due to population structure (size and aging). Nordpred separates these two components, allowing the estimation of how much of the projected mortality burden is attributable to changes in population dynamics versus underlying disease risk. We have made this distinction explicit in the methodology to avoid ambiguity.

last paragraph, line 4: “saw a decrease” and all the “saw” use: since this is a projection, saw is a stated past writting. Suggest to use something future-trend like “is estimated that…. could… might”, would be more accurate.

Authors: We thank the reviewer for this important observation. We have revised the phrasing throughout the manuscript, particularly in the final paragraph and the Results section, to ensure consistency with the projection context.

last paragraph, line 6: “ stronger increase in risk factors” which risk factors?, how did you estimated them and how did you estimated the expected change on them?

Authors: We thank the reviewer for this important observation. In response, we have revised the text to clarify that the term “risk” does not refer to specific behavioral or biological risk factors (such as obesity, smoking, or diabetes), but rather to the non-demographic component of mortality change estimated through the Nordpred model. This component reflects changes in disease incidence, treatment efficacy, and case fatality that are independent of population aging or growth. We have expanded the Methods section to provide a more precise explanation of how Nordpred separates changes in mortality into those due to population structure and those attributable to disease-related risk.

Discussion

paragraph 8 (In Peru, the…), line 10: “(refs)” and line 15 “(Ref. Pizzato et al. 2024)”: Improve structure in references

References:

add the date of consult for webpages as they tend to change.

Authors: We appreciate the reviewer’s observations. In response, we have revised the citation structure in paragraph 8 of the Discussion section to ensure consistency with the journal’s referencing style and removed placeholders such as “(refs).” We have also properly formatted the reference to Pizzato et al. (2024) according to journal guidelines. Additionally, we have updated all web-based references in the References section to include the date of access, acknowledging that online content is subject to change over time.

Tables

Table 1, title: “ cancer deaths”: why cancer deaths?

Authors: We apologize for the error. It has been corrected.

Table 1, categories: “Relative change in”: improve structure as it is confusing, and if abbreviations are used needs an explanation at the bottom of the table.

Authors: We thank the reviewer for this valuable feedback. In response, we have revised the structure of Table 1 to improve clarity and eliminate potential confusion. Specifically, we removed the “Relative change in ASR (%)” column, as its interpretation may be unclear and redundant.

Table 1, “Republica Dominicana” unify the name of the country in all the manuscript.

Authors: Done

Table 2, title: “ cancer deaths”: why cancer deaths?.

Authors: We apologize for the error. It has been corrected.

Table 2, “Number of deaths in men”: I understand this table is for women, please adjust or clarify

Authors: Done

Table 2, categories: “Relative change in”: improve structure as it is confusing, and if abbreviations are used needs an explanation at the bottom of the table.

Authors: We thank the reviewer for this valuable feedback. In response, we have revised the structure of Table 1 to improve clarity and eliminate potential confusion. Specifically, we removed the “Relative change in ASR (%)” column, as its interpretation may be unclear and redundant.

Table 2, “Republica Dominicana” unify the name of the country in all the manuscript

Authors: Done

---

## [Decision Letter · Decision Letter 1]

15 Jan 2026

Dear Dr. Torres-Roman,

Thank you for submitting your manuscript to PLOS ONE. After careful consideration, we feel that it has merit but does not fully meet PLOS ONE’s publication criteria as it currently stands. Therefore, we invite you to submit a revised version of the manuscript that addresses the points raised during the review process.

We look forward to receiving your revised manuscript.

Kind regards,

Andres Mauricio Acevedo-Melo, M.D.

Academic Editor

PLOS One

Journal Requirements:

Additional Editor Comments (if provided):

Dear Authors. Thank you for providing a revised version of your manuscript. Please proceed with minor changes before it can be considered ready for publication.

Reviewers' comments:

Reviewer's Responses to Questions

**Comments to the Author**

Reviewer #1: All comments have been addressed

Reviewer #2: All comments have been addressed

2. Is the manuscript technically sound, and do the data support the conclusions?

Reviewer #1: Yes

Reviewer #2: Yes

3. Has the statistical analysis been performed appropriately and rigorously?

Reviewer #1: Yes

Reviewer #2: Yes

4. Have the authors made all data underlying the findings in their manuscript fully available?

Reviewer #1: Yes

Reviewer #2: Yes

5. Is the manuscript presented in an intelligible fashion and written in standard English?

Reviewer #1: Yes

Reviewer #2: Yes

Reviewer #1: Thanks for addressing the methodologic concerns.

Introduction, line 4, leave an space “[3].Though”.

Put a dot at the end of 2nd paragraph of the introduction and also at the 2nd paragraph of materials and methods.

Predictions through 2035, last line, leave a space “trends[25].”

Results: start by text and at the end put “figure 1” at the end of the 1st line.

Resuls, Trends for primary hypertension, “(3.4%),Chile” needs an space, “(4.7%) El Salvador (7.7%), “ a comma is missing.

Make sure graph quality is optimal for publication.

Reviewer #2: All comments have been addressed

Authtros should be complimented because they evaluated and answered appropriately to all our comments. The work is ready now to be published and improved.

**Do you want your identity to be public for this peer review?** For information about this choice, including consent withdrawal, please see our Privacy Policy

Reviewer #1: **Yes:** Henry Becerra, MD

Reviewer #2: **Yes:** Fabrizio D'Ascenzo

---

## [Author Response · Author response to Decision Letter 2]

17 Jan 2026

Reviewer #1: Thanks for addressing the methodologic concerns.

Introduction, line 4, leave an space “[3].Though”.

Authors: Done

Put a dot at the end of 2nd paragraph of the introduction and also at the 2nd paragraph of materials and methods.

Authors: Done

Predictions through 2035, last line, leave a space “trends[25].”

Authors: Done

Results: start by text and at the end put “figure 1” at the end of the 1st line.

Authors: Done, although it would no longer be necessary to place Figure 1 at the end of the text, since each sentence has the figure, number, and letter to which it belongs.

Resuls, Trends for primary hypertension, “(3.4%),Chile” needs an space, “(4.7%) El Salvador (7.7%), “ a comma is missing.

Authors: Done

Make sure graph quality is optimal for publication.

Authors: Done

Reviewer #2: All comments have been addressed

Authtros should be complimented because they evaluated and answered appropriately to all our comments. The work is ready now to be published and improved.

Authors: We greatly appreciate the valuable comments that have significantly improved our study.

---

## [Decision Letter · Decision Letter 2]

20 Jan 2026

Differences by sex and type of hypertension in mortality from hypertensive diseases between 1997 and 2020, and predictions for 2035 in Latin American and Caribbean countries.

PONE-D-25-57467R2

Dear Dr. Torres-Roman,

We’re pleased to inform you that your manuscript has been judged scientifically suitable for publication and will be formally accepted for publication once it meets all outstanding technical requirements.

Kind regards,

Andres Mauricio Acevedo-Melo, M.D.

Academic Editor

PLOS One

Additional Editor Comments (optional):

All comments have been addressed. Please find minor issues commented as an attachment to your revised manuscript. Congratulations

Reviewers' comments:

Reviewer's Responses to Questions

**Comments to the Author**

Reviewer #1: All comments have been addressed

Reviewer #3: All comments have been addressed

2. Is the manuscript technically sound, and do the data support the conclusions?

Reviewer #1: Yes

Reviewer #3: Yes

3. Has the statistical analysis been performed appropriately and rigorously?

Reviewer #1: Yes

Reviewer #3: Yes

4. Have the authors made all data underlying the findings in their manuscript fully available?

Reviewer #1: Yes

Reviewer #3: No

5. Is the manuscript presented in an intelligible fashion and written in standard English?

Reviewer #1: Yes

Reviewer #3: Yes

Reviewer #1: All comments have been addressed, thank you.

Reviewer #3: comments are inside the document

Discuss the public health and clinical implications of the rising age‑adjusted mortality rate in hypertension, and outline recommendations for policymakers and clinicians.

To enhance the article, the discussion lacked consideration of public health policies addressing arterial hypertension and cardiovascular diseases

**Do you want your identity to be public for this peer review?** For information about this choice, including consent withdrawal, please see our Privacy Policy

Reviewer #1: **Yes:** Henry Becerra, MD

Reviewer #3: No

---

## [Editor Report · Acceptance letter]

PONE-D-25-57467R2

PLOS One

Dear Dr. Torres-Roman,

I'm pleased to inform you that your manuscript has been deemed suitable for publication in PLOS One. Congratulations! Your manuscript is now being handed over to our production team.

Kind regards,

on behalf of

Dr. Andres Mauricio Acevedo-Melo

Academic Editor

PLOS One